# Relationship between Health Counselor Characteristics and Counseling Impact on Individuals at High-Risk for Lifestyle-Related Disease: Sub-Analysis of the J-HARP Cluster-Randomized Controlled Trial

**DOI:** 10.3390/ijerph19116375

**Published:** 2022-05-24

**Authors:** Midori Noguchi, Minako Kinuta, Toshimi Sairenchi, Miyae Yamakawa, Keiko Koide, Shoko Katsura, Kazue Matsuo, Shizuko Omote, Hironori Imano, Hitoshi Nishizawa, Iichiro Shimomura, Hiroyasu Iso

**Affiliations:** 1Public Health, Department of Social Medicine, Graduate School of Medicine, Osaka University, Suita-shi 565-0871, Japan; kinuta@pbhel.med.osaka-u.ac.jp (M.K.); imano@pbhel.med.osaka-u.ac.jp (H.I.); iso@pbhel.med.osaka-u.ac.jp (H.I.); 2Department of Public Health, Okayama University Graduate School of Medicine, Dentistry and Pharmaceutical Sciences, Okayama-shi 700-8558, Japan; 3Center for Research Collaboration and Support, Comprehensive Research Facilities for Advanced Medical Science, Dokkyo Medical University, Mibu-machi 321-0293, Japan; tossair@dokkyomed.ac.jp; 4Geriatric Nursing, Division of Health Sciences, Graduate School of Medicine, Osaka University, Suita-shi 565-0871, Japan; miyatabu@sahs.med.osaka-u.ac.jp; 5Division of Health Sciences, Graduate School of Medicine, Osaka University, Suita-shi 565-0871, Japan; keiko@sahs.med.osaka-u.ac.jp; 6Graduate School of Nursing, Miyagi University, Taiwa-cho 981-3298, Japan; katura@myu.ac.jp; 7Department of Public Health Nursing, Fukuoka Jo Gakuin Nursing University, Fukuoka-shi 811-1313, Japan; ka_matsuo@fukujo.ac.jp; 8Faculty of Health Sciences, Institute of Medical, Pharmaceutical and Health Sciences, Kanazawa University, Kanazawa-shi 920-8640, Japan; omotes@mhs.mp.kanazawa-u.ac.jp; 9Department of Metabolic Medicine, Graduate School of Medicine, Osaka University, Suita-shi 565-0871, Japan; hitoshin1127@endmet.med.osaka-u.ac.jp (H.N.); ichi@endmet.med.osaka-u.ac.jp (I.S.); 10Institute for Global Health Policy Research, Bureau of International Health Cooperation, National Center for Global Health and Medicine, Tokyo 162-8655, Japan

**Keywords:** health counseling, referral to physicians, high-risk: lifestyle-related disease, community trial, public health nurse, competency

## Abstract

Early diagnosis and treatment are necessary to prevent lifestyle-related diseases among high-risk individuals. This study aimed to examine the impact of counselor characteristics on clinic visits among individuals at high risk for lifestyle-related diseases. A total of 8975 patients aged 40 to 74 years with lifestyle-related comorbidities, who underwent an annual health checkup and received health counseling, were included in this study. Data intervention timing, mode of counseling, number of counseling sessions, and explanation methods were collected. We assessed the impact of counselor characteristics, including profession (public health nurse, clinical nurse, and nutritionist), age, and years of counseling experience, on counseling outcomes. The probability ratios (95% confidence intervals) of clinic visits were 1.22 (1.11–1.35) for public health nurses and 1.04 (0.90–1.20) for nurses compared with nutritionists. After adjustment for participant and counselor characteristics and initial timing, mode, and number of counseling sessions, the corresponding probability ratios (95% confidence intervals) were 1.16 (1.05–1.29) and 1.12 (0.95–1.31), respectively. Counselor age and years of experience did not influence clinic visits of the target population. Public health nurses were more effective in increasing clinic visits among the target population owing to their profession-specific knowledge, skills, and experience.

## 1. Introduction

Increasing mortality rates due to lifestyle-related diseases are a global challenge [1]. Identification of high-risk individuals, lifestyle modifications, and prompt referral to physicians as and when required are basic strategies employed for the prevention of serious lifestyle-related diseases, such as stroke, ischemic heart disease, and chronic kidney disease [2]. High-risk individuals require medical intervention to prevent or manage disease; however, the necessary interventions are often delayed in asymptomatic cases [3]. In Japan, health counseling is provided to individuals whose health checkup findings indicate a high risk of disease with the aim of encouraging clinic visits. Nevertheless, even among the referred individuals, the uptake of clinic visits is low [4]. There are many barriers to actions for disease prevention and health promotion [5,6]. Life-style-related diseases, such as hypertension, diabetes, and hyperlipidemia, often yield no symptoms even at very high risk, making it difficult for some of the patients to take actions for visiting clinics to seek treatment under the co-payment policy [7].

Standardized health counseling for individuals at high risk of hypertension, diabetes, proteinuria, and high levels of low-density lipoprotein cholesterol (for men) helps accelerate clinic referral, as demonstrated by the Japanese trial in high-risk individuals—a nurse-led, community-based prevention program for lifestyle-related diseases [8]. In this cluster-randomized controlled trial, the cumulative proportions of clinic visits at 12 months were over 40% higher in the group receiving our original health counseling than in the group receiving standard care [9].

In Japan, public health nurses as well as clinical nurses and nutritionists may provide health counseling after annual health checkups. This study aimed to examine the impact of health counselor characteristics on the outcomes of counseling in high-risk individuals.

## 2. Materials and Methods

The present study involved conducting a secondary analysis of data from a cluster randomized clinical trial conducted in community settings (registration code: UMIN000014012). The trial recruited municipalities with >2000 participants, within the age range of 40–74 years, who underwent health checkups; all participants were covered by the national health insurance. Forty-three municipalities were randomized to 21 intervention and 22 standard care municipalities via pairwise matching randomization. The details of randomization methods and primary findings are available elsewhere [5,6].

The subjects of the present study were 10,519 individuals at a high risk for lifestyle-related disease among the 21 intervention municipalities. Risk factors included hypertension (grade II or higher: systolic blood pressure of ≥160 mmHg and/or diastolic blood pressure of ≥100 mmHg), diabetes mellitus (hemoglobin A1c levels of ≥7.0%, fasting glucose levels of ≥7.22 mmol/L, or non-fasting glucose level of ≥10.0 mmol/L), dyslipidemia (for men, low-density lipoprotein cholesterol levels of ≥4.55 mmol/L), and proteinuria (≥2+ in urinalysis). Data on risk factors were collected during the annual health checkup in 2014 and 2015. We excluded individuals aged <40 years and ≥75 years (*n* = 25) and those who had already visited physicians for conditions associated with the risk factors of interest (*n* = 1517). Finally, a total of 8977 participants were included in the analysis (Appendix A). The characteristics examined were age, professional background of counselor (public health nurse, clinical nurse, and nutritionist), and years of counseling experience for general and lifestyle-related diseases. We conducted a secondary analysis of high-risk individuals who received interventional counseling in a large cluster-randomized controlled trial to assess the impact of counselor characteristics on counseling outcomes. The number of participants assigned to public health nurses, clinical nurses, and nutritionists were 6221, 596, and 892, respectively, and among them, only 88 were men.

### 2.1. Participant Characteristics

Health checkups included questionnaires, interviews, and physical examinations involving standard methods prescribed by the Ministry of Health, Labor, and Welfare. Interviews were performed to record smoking status (non-smoker/ex-smoker/current smoker), drinking status (never/sometimes/daily), and medication use for hypertension, diabetes, and dyslipidemia.

### 2.2. Health Counselor Characteristics

Data on age, sex, and profession (public health nurse, clinical nurse, and nutritionist) of the health counselor, along with years of experience in general health counseling and years of experience in health counseling for lifestyle-related diseases were obtained using a self-administered questionnaire.

### 2.3. Health Counseling

The counseling model, known as the enhanced referral of high-risk individuals to clinics, was developed based on the health belief model [10]. The counseling sessions aim to provide high-risk individuals with information on the internal working of their bodies and future risk of lifestyle-related diseases. The model and our counseling method are de-scribed in another study [8].

During counseling sessions, health counselors used supplemental tools to explain the physiology of metabolism and mechanisms of diseases associated with high blood pressure, hyperglycemia, and hyperlipidemia (perceived susceptibility). They provided information on how blood vessels in the brain, heart, and kidneys are damaged and raised awareness on serious health problems, such as stroke, cardiovascular disorders, and renal failure, that are likely to occur if the damage is untreated (severity). These diseases are detrimental to the patient’s physical and economic life (severity). They also provided information on the potential benefits of visiting a physician (benefit) and inquired barriers to treatment, including being too busy, fear, annoyance, and lack of family support and costs (barrier). The counseled high-risk individuals were expected to make their own decisions (self-efficacy) and take the appropriate action (i.e., visiting a physician to seek further counseling and treatment). Patients were expected to receive treatment, improve their life-style habits, and continue to participate in health checkups the following year.

The counselors were trained during 2- to 3-day training sessions (seminar/workshop/group conference) held three to five times per year to dispense the knowledge, skills, and techniques required for counseling and the know-how to deal with difficult cases [8]. The counselors who were unable to participate in these training sessions received recordings for e-learning. The training sessions included the following components:(a)Mechanisms by which hypertension, hyperglycemia, and high levels of low-density lipoprotein cholesterol contribute to atherosclerosis (large vessel pathology), arteriolosclerosis (small vessel pathology), cardiovascular sclerotic arteriosclerotic diseases, and chronic kidney disease;(b)Health checkup result assessment skills to explain the pathophysiology of outcomes;(c)Need for medical treatment of hypertension, diabetes, high low-density lipoprotein cholesterol levels, and chronic kidney disease;(d)How to use information on health insurance claims;(e)Methods and implementation of health counseling based on the modified health belief model;(f)How to deal with negative responses from participants;(g)How to cooperate with primary care physicians.

In addition to the training sessions, researchers visited municipalities and provided technical support directly to the health counselors. Health counseling was monitored and judged satisfactory when the counseling record was filled properly in the items of duration time of counseling, calendar time of health checkups, mode of counseling (home visit, face-to face interview in a public place or telephone call), and the use of Appendix A.

### 2.4. Counseling Time and Mode

During the first year following the index health checkup, counseling was provided thrice: at 1–3 months, 4–6 months, and 7–9 months post health checkup. The counseling was defined as complete if the health counseling lasted over 10 min. The mode of initial health counseling was primarily a home visit and secondarily a face-to-face meeting at a municipal office or public health center; when neither of these modes was suitable, health counseling was provided over the phone.

### 2.5. Surveillance for Clinic Visits

Each municipality held insurance qualification and medical checkup data. Trained staff at each municipality office used a personal computer to link these data sets via personal identifiers to create a corresponding table for municipal and research identifiers. Staff members then deleted the municipal office identifier, name, and date of birth from the datasets and retained the research identifier and birth year and month. The data were transmitted to the central data center via registered mail. Data on health counselor characteristics and health counseling records with research identifiers were also transmitted to the central data center via registered mail, and all information was keyed into the digital dataset for analysis.

### 2.6. Statistical Analysis

The Kaplan–Meier method was used to calculate the cumulative proportions of clinic visits by high-risk individuals (hypertension, diabetes, dyslipidemia, and/or proteinuria) according to the counselor’s age (20–29, 30–39, 40–49, and ≥50 years), profession (public health nurse, clinical nurse, and nutritionist), years of experience in general counseling (<3, 3–9, 10–19, and ≥20 years), and years of experience in lifestyle disease counseling (<3, 3–5, and ≥6 years). Differences in these characteristics were evaluated using the log–rank test. Sex was not used for stratification because 98.9% of the counselors were women.

The rate ratios of cumulative proportions and their 95% confidence intervals corresponding to counselor characteristics were calculated using the Cox proportional hazard model adjusting for participants’ age, sex, smoking status (non-smokers/ex-smokers/current smokers), alcohol consumption (never/sometimes/daily), and risk factors (hypertension, diabetes mellitus, dyslipidemia for men, and proteinuria). Further adjustment was performed mutually for counselor age, years of experience in general counseling, years of experience in lifestyle disease counseling, and profession. All statistical analyses were performed using SAS 9.4 (SAS Institute Inc., Cary, NC, USA); two-tailed tests were employed, and *p*-values of <0.05 were considered statistically significant.

### 2.7. Ethical Approval and Informed Consent

Participation in the study was conducted on an opt-out method through the web-sites of all participating municipalities and Osaka University with description of the purpose and methods of the study and the option of refusing to participate. The study was approved by the Osaka University Ethics Committee (No.13237-6).

## 3. Results

Table 1 shows the characteristics of participants and counselors as well as the mode, initial timing, and number of counseling sessions according to the counselor’s profession. The mean age of the participants, proportions of grade II or higher hypertension, diabetes mellitus, and dyslipidemia did not differ significantly among the patients assigned to the three profession-related groups.

The mean age of counselors was lower for public health nurses and nutritionists than for clinical nurses. The number of years of experience in general health counseling was greatest for public health nurses, intermediate, for clinical nurses and smallest for nutritionists. The number of years of experience in lifestyle-related disease counseling was greatest for clinical nurses and intermediate for public health nurses and smallest for nutritionists. The proportion of home visits was highest for clinical nurses, intermediate for public health nurses, and lowest for nutritionists, while that of face-to-face meetings in a public place was higher for public health nurses and nutritionists than for clinical nurses. Public health nurses had the highest proportion of combined home and public face-to-face meetings. The initial timing of counseling was shorter for public health nurses and nutritionists than for clinical nurses. The number of counseling sessions did not vary significantly among the three professions.

The proportions of hypertension, diabetes mellitus, dyslipidemia, and proteinuria did not differ among the four age groups. The proportion of public health nurses was ≥85% in the 20–49-years age group and approximately 60% in the ≥50-years age groups. The numbers of years of experience in general counseling and lifestyle-related disease counseling increased with increasing age. The proportion of home visits was higher for counselors aged ≥40 years than for those aged 20–39 years. The initial timing of counseling and number of counseling sessions were similar among all age groups (Table 2).

The proportions of patients with hypertension, diabetes mellitus, dyslipidemia, and proteinuria did not differ among the groups defined by counselors’ years of experience. On comparison of the groups by years of experience, it was found that the proportions of home visits and face-to-face meetings in public places increased with counselor experience; in contrast, the proportion of incomplete counseling decreased with experience. The initial timing of counseling and number of counseling sessions were similar among the groups defined by experience (Table 3 and Appendix A).

The proportions of hypertension, diabetes mellitus, dyslipidemia, and proteinuria did not differ substantially among the groups defined by the counselors’ years of experience in lifestyle-related disease counseling (Appendix A). The proportion of public health nurses progressively decreased in groups with increasing years of experience. The proportions of home visits and face-to-face meetings were higher and lower with increasing years of experience, respectively. The initial timing of counseling and number of counseling sessions were similar among the groups defined by years of experience.

Cumulative proportions of clinic visits at 3, 6, and 12 months were consistently highest for public health nurses, intermediate for clinical nurses, and lowest for nutritionists (except for clinical nurses at 3 months) (Table 4, Figure 1). Multivariable probability ratios (95% confidence interval) of clinic visits for public health and clinical nurses were 1.22 (1.11–1.35) and 1.04 (0.90–1.20), respectively, relative to that for nutritionists (Table 4). Neither counselor age nor experience affected clinic visits (Table 4, Appendix A).

## 4. Discussion

In this sub-analysis of a large clinical trial dataset, counselors who were public health nurses achieved a higher cumulative proportion of clinic visits than their counterparts. Neither counselor age nor experience affected clinic visits.

The Quad Council Coalition list of major competency categories for public health nurses included the following eight categories: (1) assessment and analytic skills, (2) policy development/program planning skills, (3) communication skills, (4) cultural competency skills, (5) community dimensions skills, (6) public health science skills, (7) financial planning, evaluation, and management skills, and (8) leadership and systems thinking skills [11]. Communication skills included critical thinking and complex decision-making skills, adapting to individuals, families, workplaces, and communities. The eight competency categories could contribute to acquire the sufficient information on the counselee’s demographics, including age, sex, place of residence, occupation, family composition, and socioeconomic status, which can affect health behaviors [12]. Public health nurses in Japan are responsible for public health activities through home visits to residents and are trained to identify key individual health behaviors, such as diet, smoking and drinking status, clothing, and household environment during face-to-face conversations to improve their health behaviors [13].

On the other hand, the competencies of clinical nurses are (1) embodying a helping role, (2) teaching and coaching, (3) diagnostic functions, (4) managing situations, (5) therapeutic interventions, (6) quality control, and (7) undertaking a work role in the context of care provided to individual patients [14]. Namely, the competencies focus more on taking care of individual patients and less on perspectives of disease prevention and health promotion, which contrast with those of public health nurses.

According to the WHO’s five-dimensional adherence model [15,16], the factors related to adherence are socioeconomic, healthcare system, condition-related, therapy-related, and patient-related factors. In this study, we focused on one of the health care system factors, i.e., competency of health providers as the exposure variable. As for confounding variables, we used the demographic variables (age and sex) as socioeconomic factors and comorbidities (hypertension, diabetes mellitus, dyslipidemia, and proteinuria) as condition-related factors. Therapy-related factors were not considered because the participants were not treated before the counseling. Patient-related factors such as knowledge and self-efficacy were regarded as intermediate variables between health counseling and the outcome of clinical visits.

The present finding that public health nurses had a higher proportion of home visits and face-to-face sessions in public places than their counterparts is an indication of their expertise. Although the expected role of public health nurses varies by country [17,18], our results indicate the effectiveness of public health nurses at preventing lifestyle-related disease by encouraging clinic visits. Since clinical diagnosis and medical treatment for asymptomatic lifestyle-related diseases cause the patients to pay copayment [7], appropriate counseling is needed, especially for people with low income or economic burden due to health conditions of family members.

The strength of our study is its large sample size, which enables to evaluate differences in outcomes among different professional groups under the same intervention program. There are several limitations to our study. First, the clinical nurses were, on average, over 10 years older than public health nurses and nutritionists, which made it difficult to adjust for age sufficiently. Second, unmeasured factors, such as socioeconomic factors, other than demographic variables, such as education and income, as well as residual confounding factors may affect the present findings. Third, most of the counselors in our study were women, so our findings are not generalizable to male counselors. The proportion of men among public health nurses was only 1.9% in Japan [19].

## 5. Conclusions

The highest cumulative proportions of clinic visits among high-risk individuals were achieved by public health nurses, followed by clinical nurses and nutritionists, supporting the concept that public health nurses’ professional knowledge, skills, and experiences may improve the effectiveness of counseling. Public health nurses are expected to have a crucial role to accelerate clinical visits for high-risk individuals in the prevention of serious lifestyle-related diseases, such as stroke, ischemic heart disease, and chronic kidney disease.

## Figures and Tables

**Figure 1 ijerph-19-06375-f001:**
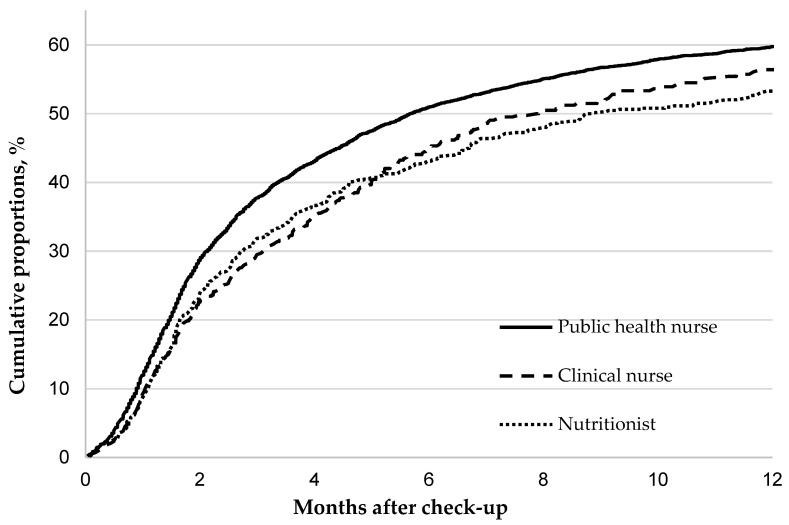
The cumulative proportions of clinic visits for participants according to health counselor’s profession.

**Table 1 ijerph-19-06375-t001:** Characteristics of the participants and counselors according to the counselors’ profession.

Profession	Public Health Nurse		Clinical Nurse		Nutritionist	
**Characteristics of participants**						
Number of participants	6219		596		892	
Age, years, mean ± SD	63.1	±8.5	63.8	±8.2	64.4	±7.9
Men, *n* (%)	4139	(66.6)	352	(59.1)	583	(65.4)
Grade II or higher hypertension, *n* (%)	3619	(58.2)	338	(56.7)	520	(58.3)
Diabetes mellitus, *n* (%)	1184	(19.0)	119	(20.0)	161	(18.1)
Dyslipidemia among men, *n* (%)	1410	(34.3)	107	(30.6)	220	(38.1)
Proteinuria, *n* (%)	606	(9.8)	91	(15.4)	92	(10.3)
**Characteristics of counselors**						
Women, *n* (%)	6142	(98.8)	596	(100.0)	881	(98.8)
Men, *n* (%)	77	(1.2)	0	(0.0)	11	(1.2)
Age, years, mean ± SD	39.5	±9.6	51.7	±6.4	39.4	±9.4
Years of experience in general counseling, mean ± SD	12.2	±10.1	9.9	±6.6	6.5	±8.3
<3, *n* (%)	1511	(24.3)	92	(15.4)	456	(51.1)
3–9, *n* (%)	1629	(26.2)	251	(42.1)	274	(30.7)
10–19, *n* (%)	1466	(23.6)	224	(37.6)	87	(9.8)
≥20, *n* (%)	1611	(25.9)	29	(4.9)	75	(8.4)
Years of experience for lifestyle-related disease counseling, mean ± SD	4.3	±5.3	8.7	±6.1	3.4	±3.6
<3, *n* (%)	3418	(55.0)	111	(18.6)	503	(56.4)
3–5, *n* (%)	1493	(24.0)	87	(14.6)	311	(34.9)
≥6, *n* (%)	1303	(21.0)	398	(66.8)	78	(8.7)
Counseling mode, *n* (%)						
Home visit	3375	(54.3)	418	(70.1)	402	(45.1)
Face-to-face in a public place	1694	(27.2)	33	(5.5)	281	(31.5)
Telephone	307	(4.9)	33	(5.5)	36	(4.0)
Incomplete	845	(13.6)	112	(18.8)	173	(19.4)
Initial timing, *n* (%)						
≤45 days	2489	(40.0)	42	(7.1)	287	(32.2)
46–90 days	1688	(27.1)	109	(18.3)	257	(28.8)
≥91 days	1335	(21.5)	337	(56.5)	189	(21.2)
Incomplete	707	(11.4)	108	(18.1)	159	(17.8)
Number of counseling sessions, *n* (%)						
1	2566	(41.3)	250	(42.0)	332	(37.2)
2	1753	(28.2)	205	(34.4)	223	(25.0)
3	1250	(20.1)	47	(7.9)	182	(20.4)
Incomplete	650	(10.5)	94	(15.8)	155	(17.4)

SD, standard deviation.

**Table 2 ijerph-19-06375-t002:** Characteristics the participants and counselors according to the age of the counselors.

Ages of Health Counselors, Years	20–29		30–39		40–49		≥50	
**Characteristics participants**								
Number of participants	1336		2385		2327		1643	
Age, years, mean ± SD	63.3	±8.2	63.6	±8.3	63.1	±8.5	63.1	±8.5
Men, *n* (%)	873	(65.3)	1563	(65.5)	1553	(66.7)	1077	(65.6)
Grade II or higher hypertension, *n* (%)	788	(58.9)	1391	(58.3)	1346	(57.8)	942	(57.3)
Diabetes mellitus, *n* (%)	239	(17.9)	450	(18.9)	457	(19.7)	314	(19.1)
Dyslipidemia among men, (%)	310	(35.6)	533	(34.4)	531	(34.5)	363	(33.9)
Proteinuria, *n* (%)	143	(10.7)	241	(10.1)	215	(9.3)	184	(11.2)
**Characteristics of counselors**								
Women, *n* (%)	1321	(98.9)	2335	(97.9)	2315	(99.5)	1632	(99.3)
Profession								
Public health, *n* (%)	1171	(87.7)	2021	(84.7)	1996	(85.8)	1031	(62.8)
Clinical nurse, *n* (%)	0	(0.0)	37	(1.6)	157	(6.8)	386	(23.5)
Nutritionist, *n* (%)	165	(12.3)	327	(13.7)	174	(7.5)	226	(13.8)
Years of experience in general counseling, mean ± SD	2.9	±2.1	7.5	±4.8	14.9	±9.0	18.9	±12.1
<3, *n* (%)	791	(59.2)	561	(23.5)	432	(18.6)	275	(16.8)
3–9, *n* (%)	545	(40.8)	1055	(44.2)	310	(13.3)	244	(14.9)
10–19, *n* (%)	0	(0.0)	769	(32.2)	695	(29.9)	297	(18.1)
≥20, *n* (%)	0	(0.0)	0	(0.0)	890	(38.3)	825	(50.3)
Years of experience for lifestyle-related disease counseling, mean ± SD	2.1	±1.5	3.7	±3.2	4.9	±5.6	7.1	±7.8
<3, *n* (%)	981	(73.7)	1285	(53.9)	1092	(47.0)	674	(41.0)
3–5, *n* (%)	326	(24.5)	647	(27.1)	652	(28.0)	266	(16.2)
≥6, *n* (%)	25	(1.9)	453	(19.0)	582	(25.0)	703	(42.8)
Counseling mode, *n* (%)								
Home visit	641	(48.0)	1144	(48.0)	1411	(60.6)	988	(60.1)
Face-to-face in a public place	421	(31.5)	679	(28.5)	513	(22.1)	394	(24.0)
Telephone	78	(5.8)	133	(5.6)	90	(3.9)	72	(4.4)
Incomplete	196	(14.7)	429	(18.0)	313	(13.5)	189	(11.5)
Initial timing, *n* (%)								
≤45 days	474	(35.5)	840	(35.2)	974	(41.9)	529	(32.2)
46–90 days	416	(31.1)	672	(28.2)	542	(23.3)	420	(25.6)
≥91 days	278	(20.8)	484	(20.3)	550	(23.6)	541	(32.9)
Incomplete	168	(12.6)	389	(16.3)	261	(11.2)	153	(9.3)
Number of counseling sessions, *n* (%)								
1	556	(41.6)	1024	(42.9)	979	(42.1)	583	(35.5)
2	323	(24.2)	603	(25.3)	714	(30.7)	534	(32.5)
3	289	(21.6)	404	(16.9)	400	(17.2)	386	(23.5)
Incomplete	168	(12.6)	354	(14.8)	234	(10.1)	140	(8.5)

SD, standard deviation.

**Table 3 ijerph-19-06375-t003:** Characteristics of the participants and counselors according to the experience in general counseling.

Years of Experience	<3		3–9		10–19		≥20	
**Characteristics of participants**								
Number of participants	2059		2154		1777		1715	
Age, years, mean ± SD	64.1	±8.1	63.3	±8.5	63.3	±8.2	62.4	±8.7
Men, *n* (%)	1331	(64.6)	1431	(66.4)	1167	(65.7)	1144	(66.7)
Grade II or higher hypertension, *n* (%)	1237	(60.1)	1225	(56.9)	1002	(56.4)	1012	(59.0)
Diabetes mellitus, *n* (%)	367	(17.8)	394	(18.3)	396	(22.3)	307	(17.9)
Dyslipidemia among men, *n* (%)	463	(34.9)	520	(36.5)	368	(31.7)	385	(34.0)
Proteinuria, *n* (%)	199	(9.7)	236	(11.0)	197	(11.1)	157	(9.2)
**Characteristics of counselors**								
Women, *n* (%)	2005	(97.4)	2154	(100.0)	1743	(98.1)	1715	(100.0)
Profession								
Public health, *n* (%)	1511	(73.4)	1629	(75.6)	1466	(82.5)	1611	(93.9)
Clinical nurse, *n* (%)	92	(4.5)	251	(11.7)	224	(12.6)	29	(1.7)
Nutritionist, *n* (%)	456	(22.2)	274	(12.7)	87	(4.9)	75	(4.4)
Years of experience for lifestyle-related disease counseling, mean ± SD	1.3	±0.8	4.5	±2.3	6.1	±5.0	6.9	±8.6
<3, *n* (%)	2050	(99.8)	559	(26.0)	674	(38.0)	747	(43.6)
3–5, *n* (%)	5	(0.2)	1021	(47.4)	416	(23.4)	449	(26.2)
≥6, *n* (%)	0	(0.0)	574	(26.7)	686	(38.6)	519	(30.3)
Counseling mode, *n* (%)								
Home visit	1045	(50.8)	1110	(51.5)	1019	(57.3)	1021	(59.5)
Face-to-face in a public place	510	(24.8)	539	(25.0)	468	(26.3)	490	(28.6)
Telephone	120	(5.8)	142	(6.6)	53	(3.0)	59	(3.4)
Incomplete	384	(18.7)	363	(16.9)	237	(13.3)	145	(8.5)
Initial timing, *n* (%)								
≤45 days	620	(30.1)	609	(28.3)	724	(40.7)	865	(50.4)
46–90 days	574	(27.9)	613	(28.5)	418	(23.5)	449	(26.2)
≥91 days	538	(26.1)	602	(28.0)	427	(24.0)	293	(17.1)
Incomplete	327	(15.9)	330	(15.3)	208	(11.7)	108	(6.3)
Number of counseling sessions, *n* (%)								
1	829	(40.3)	991	(46.0)	638	(35.9)	689	(40.2)
2	554	(26.9)	554	(25.7)	561	(31.6)	512	(29.9)
3	352	(17.1)	317	(14.7)	393	(22.1)	417	(24.3)
Incomplete	324	(15.7)	292	(13.6)	185	(10.4)	97	(5.7)

SD, standard deviation.

**Table 4 ijerph-19-06375-t004:** Cumulative proportions of clinic visits and probability ratios (95% confidence intervals) according to the counselors’ profession, age of counselors, years of experience for general counseling, and years of experience for lifestyle-related counseling.

Profession	Public Health Nurse		Clinical Nurse		Nutritionist		*p*-Value		
No. at risk	6219		596		892				
No. of clinic visits	3439		302		438				
Cumulative proportion of clinic visits (95% CI)									
3 months	37.8	(36.6–39.0)	29.5	(26.0–33.4)	31.9	(28.9–35.0)	<0.001		
6 months	50.9	(49.7–52.2)	45.1	(41.1–49.3)	43.1	(39.9–46.5)	<0.001		
12 months	59.8	(58.4–61.1)	56.4	(52.0–60.9)	53.3	(49.8–56.9)	<0.001		
Probability ratio (95% Cl)	1.22	(1.11–1.35)	1.04	(0.90–1.20)	1.00				
Multivariable probability ratio (95% Cl)									
Model 1 ^a^	1.18	(1.07–1.31)	0.99	(0.85–1.16)	1.00				
Model 2 ^b^	1.16	(1.05–1.29)	1.12	(0.95–1.31)	1.00				
**Ages of counselors**	**20–29**		**30–39**		**40–49**		**≥50**		***p*-value**
No. at risk	1336		2385		2327		1643		
No. of clinic visits	739		1230		1290		912		
Cumulative proportion of clinic visits (95% CI)									
3 months	37.1	(34.6–39.8)	34.5	(32.7–36.5)	38.2	(36.2–40.2)	36.3	(34.1–38.7)	0.005
6 months	49.8	(47.1–52.6)	47.5	(45.5–49.6)	50.9	(48.9–53.0)	50.6	(48.1–53.1)	<0.001
12 months	60.6	(57.7–63.5)	56.4	(54.2–58.6)	60.0	(57.8–62.1)	58.9	(56.4–61.5)	<0.001
Probability ratio (95% CI)	1.00		0.91	(0.83–1.00)	1.01	(0.92–1.11)	0.97	(0.88–1.07)	
Multivariable probability ratio (95% Cl)									
Model 1 ^a^	1.00		0.87	(0.79–0.96)	0.89	(0.80–0.99)	0.86	(0.76–0.98)	
Model 2 ^b^	1.00		0.90	(0.82–0.99)	0.91	(0.82–1.02)	0.88	(0.77–1.00)	
**Years of experience for general counseling**	**<3**		**3–9**		**10–19**		**≥20**		***p*-value**
No at risk	2059		2154		1777		1715		
No. of clinic visits	1064		1106		1007		1002		
Cumulative proportion of clinic visits (95% CI)									
3 months	33.2	(31.2–35.3)	33.9	(31.9–35.9)	38.3	(36.0–40.6)	41.6	(39.3–44.0)	<0.001
6 months	46.6	(44.4–48.8)	47.2	(45.0–49.4)	52.0	(49.6–54.4)	53.7	(51.3–56.1)	<0.001
12 months	57.0	(54.6–59.4)	56.0	(53.8–58.4)	60.9	(58.4–63.3)	62.1	(59.6–64.6)	<0.001
Probability ratio (95% CI)	1.00		0.99	(0.91–1.08)	1.13	(1.04–1.23)	1.20	(1.10–1.31)	
Multivariable probability ratio (95% Cl)									
Model 1 ^a^	1.00		0.97	(0.89–1.06)	1.09	(0.99–1.20)	1.21	(1.08–1.35)	
Model 2 ^b^	1.00		0.97	(0.89–1.06)	1.02	(0.92–1.12)	1.07	(0.96–1.21)	
**Years of experience for lifestyle-related disease counseling**	**<3**		**3–5**		**≥6**		***p*-value**		
No. at risk	4032		1891		1779				
No. of clinic visits	2198		1020		958				
Cumulative proportion of clinic visits (95% CI)									
3 months	36.1	(34.6–37.6)	37.1	(34.9–39.3)	36.5	(34.3–38.8)	0.037		
6 months	49.4	(47.9–51.0)	49.5	(47.2–51.8)	50.0	(47.6–52.4)	<0.001		
12 months	59.0	(57.3–60.6)	58.4	(56.0–60.8)	58.6	(56.1–61.2)	<0.001		
Probability ratio (95% CI)	1.00		1.00	(0.93–1.08)	0.99	(0.92–1.07)			
Multivariable probability ratio (95% Cl)									
Model 1 ^a^	1.00		0.98	(0.91–1.06)	0.96	(0.88–1.04)			
Model 2 ^b^	1.00		0.98	(0.91–1.06)	0.95	(0.88–1.04)			

^a^ Adjusted for participant’s age, sex, smoking status, drinking status, and risk factors, and adjusted mutually for the other exposure variables such as profession, age of counselors, years of experience for general counseling, and years of experience for lifestyle-related disease counseling. ^b^ Adjusted further for the initial timing, the mode, and the number of counseling.

## Data Availability

On reasonable request, derived data supporting the findings of this study are available from the corresponding author after approval from the Ethical Committee of the Osaka University.

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
