# Peer review of "Relationship between Health Counselor Characteristics and Counseling Impact on Individuals at High-Risk for Lifestyle-Related Disease: Sub-Analysis of the J-HARP Cluster-Randomized Controlled Trial"

_ijerph, 2022, doi:10.3390/ijerph19116375_

Round 1

Reviewer 1 Report

The work, Relationship between health counselor characteristics and counseling impact on Individuals at high-risk for lifestyle-related disease: Sub-analysis of the J-HARP cluster-randomized controlled trial, by Noguchi et al. presents a very interesting issue. However, it requires a few editorial modifications and changes to the text content.

In section 2.3 on line 135-147, you only need to use the dots as bullets or letter. Please, choose one type of bullet point.

All tables (1, 2, 3 and 4) presented in the text, as well as tables from additional materials S2 require editorial changes. Please remove the spaces between the figures and brackets and between the mean value and SD, because in its current form it is very difficult to read and analyze the results you provide.

The whole text requires justification, especially the sections placed on lines: 195-214; 217-223; 225-238; 252-293; 301-306.

The presented graphs (Figure 1 from the main text) and the figures from the additional materials (S3, S4 and S5) require a lot of work. First of all, their quality is very poor and they look as if they were drawn in Paint, not generated by, for example, Excell. I am asking you to correct them.

I am asking you to correctly quote the materials attached to the supplemantary in the main text. In the current situation they are quoted once as Additional file 2 and once as Supplemantary file 3. Please quote them exactly, eg Table S2 etc.

The Conclusion part is not enough. I am asking you to expand it and refer to the publication goal set at the beginning of the work and to consider how your research relates to society.

I would also like to ask if there is a possibility of extending part of the discussion with a larger number of cited articles? Due to the nature of the work, this may not be possible, but it would be good to expand on this part as well.

Reviewer 2 Report

Dear author,

Regardless of what you mention in your manuscript as a strength of the large sample size and this is a substudy of a large clinical trial dataset. From my point of view, this manuscript is lacking in quality and consistency, mainly, in the method and discussion sections. These sections are inconsistent and confusing as I try to explain here: 

Introduction:
In this section, I recommend you end with the aim of your study and I suggest changing the paragraph from lines 66 to 70, for the next material and method section. Moreover, in the paragraph from lines 71 to 72  I suggest changing to the discussion section in order to support some ideas offered.

Materials and methods section:
In this section, I recommend you delete duplicated information offered in lines 78 and 87 according to "the among the 21 intervention municipalities" of your study. It is duplicated information.
Furthermore, there is extra information in this section. I do not understand why is relevant some information shown in the 2.3. Health counseling point. Besides, from my point of view, this is not a counseling model, if not a clinic visit/probability ratio model. I think that you show the information from a large clinical trial dataset but some of them are not relevant for this manuscript.
Furthermore, in the statement shown in line 109, I would like to ask if only patient information can affect health behaviors? 
Moreover, in the paragraph from lines 122 to 124, please explain how was it investigated? or add a reference cite to justify it.
personally, I would like to ask why only use the risk factors mentioned in the paragraph of line 180 in your study? As a reviewer and reader of your manuscript, I have this doubt that I consider should be explained in this section.
Additionally, there is a lack of information on the Ethics and Informed Consent Statements in this section. Please, add it to your manuscript.
Finally, please use the same space format when you use these kinds of symbols (>/<) in your text.

Results 
I recommend you change and add the sentence from lines 189 to 190 to the previous material and method section.
Please, correct the word "profession" for the correct one in the title of the table 1 description, use a correct text format for the text of all tables, without spaces between symbols and text, and order in the correct place.

Discussion:
Please, justify by a reference cite at the end of the sentence of line 262. 
In this section, I cannot understand what is the value of the third and fourth paragraphs. I think have no sense according to the aim and results are shown in your manuscript. I insist, that you are not doing a not a counseling model, if not a clinic visit/probability ratio model. Please adapt your discussion according to it.
Regarding the low number of references used in your manuscript, I suggest you add some references to support some of your statements. 
In line 296, I suggest using the connector "such as" to introduce all the limitations of your study.
Please, add as a limitation that almost all the counselors analyzed in your study were women.

Finally, I would like to recommend you justify all text formats in your manuscript

Yours faithfully,

Round 2

Reviewer 1 Report

Thank you very much for your response and for making all the changes I have requested. I only have one more remark about the figures presented, both in the main text and in additional materials - is it possible to reduce the scale in the charts (of course, it is standardized in all charts), so that the lines presented there are further apart (everything overlaps in places ). This would allow for better legibility. Or maybe different line colors can be used? In its present form, it is hard to tell if it is a broken line, dotted line, or solid thin line.

Reviewer 2 Report

Dear authors,

I consider that several of my comments and suggestions had been answered point-by-point. However, there are some aspects that have not been corrected yet. 
Please, use the same space format when you use these kinds of symbols (>/<) in your text.
Please, correct the word "nerse" for the correct one in figure 1.
Furthermore, I continue to have some doubts that from my point of view should be improved in the 2.3. Health counseling point.
Please, add a reference cite at the end of the paragraph ended in line 128, ) if it is possible, and try to justify how was measured the satisfactory level of Health counseling mentioned in line 147.
Moreover, I cannot understand why there is not a table with the characteristics of the participants and counselors according to their years of experience in lifestyle disease counseling, as the rest of the outcomes measured.
Finally, and from my point of view as a mandatory limitation, you have to increase your second limitation. You have mentioned that unmeasured factors such as socioeconomic, status and personality and residual confounding factors may affect the present findings. However, I would like to suggest you again mention the different dimensions and levels that should manage to improve the effectiveness of counseling impact. For instance, according to the WHO's five dimensions model, you should specify all these: Socioeconomic factors, healthcare system factors, condition-related factors, therapy-related factors, and patient-related factors (suggested cite reference: Calonge Pascual S, Casajús Mallén JA, González-Gross M. Adherence Factors Related to Exercise Prescriptions in Healthcare Settings: A Review of the Scientific Literature. Res Q Exerc Sport. 2020 Sep 9:1-10. doi: 10.1080/02701367.2020.1788699). Furthermore, according to a social-ecological approach you can consider all these levels: Individual, interpersonal, organizational, community and policy, to improve the effectiveness of counseling (suggested cite reference: Burke, S., Utley, A., Belchamber, C., & McDowall, L. (2020). Physical Activity in Hospice Care: A Social Ecological Perspective to Inform Policy and Practice. Res Q Exerc Sport, 1-14. doi: 10.1080/02701367.2019.1687808). I consider it would add scientific value to your manuscript and complete your discussion section with new arguments.

In the present form, I cannot accept the manuscript to be published.

Yours faithfully,
